# Social and Nutritional Profiles of Pregnant Women: A Cluster Analysis on the “MAMI-MED” Cohort

**DOI:** 10.3390/nu16233975

**Published:** 2024-11-21

**Authors:** Giuliana Favara, Andrea Maugeri, Martina Barchitta, Roberta Magnano San Lio, Maria Clara La Rosa, Claudia La Mastra, Fabiola Galvani, Elisa Pappalardo, Carla Ettore, Giuseppe Ettore, Antonella Agodi

**Affiliations:** 1Department of Medical and Surgical Sciences and Advanced Technologies “GF Ingrassia”, University of Catania, 95123 Catania, Italy; giuliana.favara@unict.it (G.F.); andrea.maugeri@unict.it (A.M.); martina.barchitta@unict.it (M.B.); robertamagnanosanlio@unict.it (R.M.S.L.); mariclalarosa@gmail.com (M.C.L.R.); claudia.lamastra@outlook.it (C.L.M.); 2Department of Obstetrics and Gynaecology, Azienda di Rilievo Nazionale e di Alta Specializzazione (ARNAS) Garibaldi Nesima, 95124 Catania, Italy; fabiolagalvani38@gmail.com (F.G.); elypappalardo@yahoo.it (E.P.); carla.ettore@hotmail.it (C.E.); giuseppe.ettore@gmail.com (G.E.)

**Keywords:** neonatal outcomes, nutrition, pregnancy, maternal diet, educational level, social determinants

## Abstract

Background/Objectives: During the pre-conceptional period, addressing social determinants of health (SDOH) is essential for reducing maternal health disparities, particularly among disadvantaged groups. Key SDOH factors such as income, education, and healthcare access significantly influence maternal and infant outcomes, increasing risks like miscarriage, preterm birth, and pregnancy complications. Here, we aimed to explore maternal and neonatal characteristics according to socio-economic status. Thus, we identified clusters of pregnant women with similar social and behavioral characteristics and explored their variability in terms of neonatal outcomes. Methods: Data from 1512 pregnant women in the “MAMI-MED” cohort at ARNAS Garibaldi Nesima in Catania were analyzed. A two-step cluster analysis grouped the women based on education level, employment status, pre-pregnancy nutritional status, and Mediterranean diet score (MDS). Results: Two clusters of pregnant women were identified. Cluster 1 (n = 739) consisted of women with lower educational attainment who were unemployed, overweight and/or obese, and had a lower mean MDS. Instead, cluster 2 (n = 773) was mostly characterized by women with a medium–high level of education who were employed, had normal weight, and had a higher average MDS. Women in cluster 1 had significantly higher proportions of preterm births (*p* = 0.004), low-birth weight newborns (*p* = 0.002), and large-for-gestational-age newborns. Differences in gestational week (*p* < 0.001), birth weight (*p* < 0.001), and newborn length (*p* = 0.004) were also noted between the two clusters. Conclusions: Cluster analysis can help identify high-risk groups who may benefit from personalized public health interventions. Our results highlight the need to examine the complex interactions between socio-demographic, behavioral, and genetic factors that contribute to maternal–infant health.

## 1. Introduction

During the pre-conceptional period, addressing social determinants responsible for shaping maternal and infant health trajectories requires sustained and personalized interventions to reduce the burden of maternal health disparities, especially among socially disadvantaged groups [1,2]. Social determinants of health (SDOH) play a pivotal role in shaping disparities in maternal and infant health outcomes. Key factors include socioeconomic conditions such as income, education, employment, and access to healthcare, along with community dynamics and the physical environment. These elements can significantly influence reproductive health, heightening the risk of adverse outcomes like miscarriage, preterm birth, and preeclampsia and contributing to broader pregnancy complications [3]. The peri-conceptional environment is influenced by a combination of maternal pre-existing medical conditions and modifiable lifestyle factors. During this critical period, maternal behaviors and lifestyle choices play a pivotal role in the healthy development of pregnancy, with the potential to significantly impact both maternal and fetal outcomes [4,5,6]. The concept of “nutritional programming” underscores the significant influence of optimal maternal nutrition and appropriate weight gain on health outcomes, both in the short term during intrauterine development and in the long term throughout life [7,8,9,10,11]. In this context, the level of nutritional knowledge, personal and clinical characteristics, as well as socio-economic and cultural factors, may be considered responsible for woman’s lifestyles and eating choices [7,12,13,14,15]. Indeed, existing evidence suggested that women with lower socio-economic level often follow unhealthy behaviors, such as smoking and a sedentary life, and make poor dietary choices characterized by lower consumption of fruit and vegetables and greater consumption of processed meet, salty meals, refined grains, and added fats [16]. Thus, women who follow unhealthy behaviors are exposed to a higher risk of pregnancy complications—such as anemia, gestational hypertension, and gestational diabetes—and can adversely affect fetal development (i.e., low birth weight (LBW), preterm birth (PTB)), contributing to other chronic health conditions that may persist into childhood and beyond [16,17,18,19,20,21]. Despite advancements in promoting women’s health, achieving equality in social and health outcomes for pregnant individuals remains challenging. To develop effective interventions that reduce these disparities and improve pregnancy outcomes, it is crucial to understand the underlying mediators or causes of these observed differences [3]. In this context, the purpose of the present work was to evaluate how maternal dietary habits and socio-economic characteristics can affect the risk of adverse maternal and child health outcomes. Specifically, we identified clusters of pregnant women with similar social and behavioral characteristics and explored their variability in terms of neonatal outcomes.

## 2. Materials and Methods

### 2.1. Study Design

The “MAMI-MED” cohort is a prospective study launched in 2020, designed to recruit pregnant women during their first trimester visit at the “Azienda di Rilievo Nazionale e di Alta Specializzazione (ARNAS) Garibaldi Nesima” in Catania, Italy. This study aimed to investigate the impact of the exposome on the health of mother–child dyads, utilizing protocols and methodologies closely aligned with those of the “Mamma and Bambino” cohort, which has been ongoing in Catania since 2015 [22,23,24,25,26,27,28]. The study plan included interviews at recruitment, at delivery, and after 12, 24, and 48 months to collect information regarding the health, socio-economic status, and lifestyles of the parents, as well as data concerning the health, habits, and diet of the child. The MAMI-MED study protocol was approved by the ethics committee “Catania 2” (Protocol numbers: 487/CE, 71/2020/CECT2; 157/CEL) and is in line with the Declaration of Helsinki. All women were fully informed about the study purpose and procedures and signed their informed consent to participate.

### 2.2. Data Collection

This analysis focused on mother–child dyads from the “MAMI-MED” cohort with comprehensive data on socio-demographic factors, dietary patterns, and birth outcomes. During the recruitment phase, detailed information about the women’s socio-demographic background, lifestyle choices, and health conditions was collected using a structured questionnaire. Maternal level of education was categorized into two levels, as follows: low educational level if women had ≤8 years of school and medium–high level if women had >8 years of school. Mothers were categorized as either employed or unemployed, with the latter group including students and homemakers. Pre-gestational nutritional status was classified into four categories: underweight, normal weight, overweight, and obese. Data on the week of delivery and child health outcomes were obtained through tailored questionnaires administered at birth. Key neonatal outcomes included preterm birth (PTB), defined as spontaneous delivery before 37 weeks of gestation, and birth weight relative to gestational age, categorized as small (SGA), adequate (AGA), or large (LGA) for gestational age based on sex-specific national reference charts [29]. Additional neonatal measurements included birth weight and length.

### 2.3. Dietary Assessment

At recruitment, a 95-item semi-quantitative food frequency questionnaire (FFQ) was administered by trained epidemiologists to gather maternal dietary information for the month preceding recruitment [28]. Women were asked to report their frequency of food consumption, categorized into 12 levels ranging from “almost never” to “two or more times a day”, along with the serving size, classified as small, medium, or large. Medium serving sizes were standardized based on typical weight or volume measures commonly consumed in the Italian population, while small and large servings were defined as half or 1.5 times (or more) the medium size, respectively. To enhance accuracy and minimize reporting errors, participants were provided with a detailed photographic atlas to visually estimate portion sizes. Daily intake for each food group was then calculated by multiplying the reported consumption frequency by the corresponding portion size by using the U.S. Department of Agriculture (USDA) Food Composition Database (https://fdc.nal.usda.gov/, accessed on 1 July 2024) adapted to typical Italian foods [30]. Adherence to the Mediterranean diet (MD) was assessed using the Mediterranean diet score (MDS) [31,32], which assesses dietary patterns across nine food categories: fruits and nuts, vegetables, legumes, cereals, lipids, fish, dairy products, meat products, alcohol, and the ratio of unsaturated to saturated fats. For vegetables, legumes, fruits and nuts, cereals, fish, and the unsaturated-to-saturated fat ratio, women were assigned a score of 0 if their intake was at or below the population median and a score of 1 if their intake exceeded the median. Conversely, for dairy and meat products, a score of 1 was assigned to those consuming below the median and a score of 0 to those above. Regarding alcohol, a score of 1 was assigned to women consuming between 5 and 25 g per day. Consequently, the MDS ranged from 0 to 9 and was categorized as follows: low adherence (MDS range: 0–3), medium adherence (MDS range: 4–6), or high adherence (MDS range: 7–9). 

### 2.4. Cluster Analysis

A cluster analysis was employed to uncover natural groupings that are not readily apparent within the dataset of the MAMI-MED cohort. This method aims to achieve high similarity within clusters and significant variability between clusters. The two-step cluster analysis method was used to identify distinct groups of pregnant women based on their level of education, employment status, pre-gestational nutritional status, and MDS. This method involves constructing a cluster feature (CF) tree, starting with the first case at the root and adding each subsequent case to an existing node or forming a new node based on their similarity. An agglomerative clustering algorithm, capable of handling both categorical and continuous variables, is then used to categorize the CF tree’s nodes and automatically generate a range of clustering options. The determination of the optimal number of clusters was automated using Schwarz’s Bayesian information criterion (SBIC), which identifies the most suitable model by maximizing the likelihood function. The two-step cluster analysis applied the log-likelihood distance measure, enabling the integration of both categorical and continuous variables while assuming independence among variables within the cluster model. Variables were assessed for their predictive importance, with scores ranging from 0.1 (indicating minimal predictive relevance) to 1.0 (indicating maximum predictive relevance). To enhance the reliability of the clustering solution, variables with predictive importance scores below 0.2 were excluded, and the analysis was re-evaluated to confirm consistency and validity [33,34].

### 2.5. Statistical Analysis

Statistical analyses were conducted using SPSS software version 26.0 (IBM Corp., Armonk, NY, USA). Descriptive statistics, including the median and interquartile range (IQR) or percentage, were used to characterize the population. The Kolmogorov–Smirnov test assessed the normal distribution of variables. For categorical variables, the chi-squared test was employed, focusing on the right tail of the distribution, while the Kruskal–Wallis test was used for continuous variables. All statistical tests were conducted with a significance level set at *p* < 0.05.

## 3. Results

### 3.1. Study Sample

The present analysis included 1512 women—enrolled in the “MAMI-MED” cohort—who completed pregnancy at a median week of delivery of 39 (IQR = 2). Women reported a median age of 31 years (IQR = 7), and about half (54.4%) were primiparous. Regarding socioeconomic status, 75.2% of the women reported having a high-medium level of education, and 51.6% were employed. In terms of lifestyles during pregnancy, 90.3% reported not being smoker, 62.1% reported medium–high adherence to MD, and 84.8% did not follow dietary restrictions, with a median total energy intake of 1714 kcal (IQR = 564). Before pregnancy, 58.3% of women were normal weight and reported a median body mass index (BMI) of 23.4 kg/m^2^ (IQR = 6.1). According to the gestational weight gain (GWG; median = 11 kg; IQR = 7), 39.5% of women reported a reduced weight gain, 28.2% reported an excessive weight gain, and 32.3% reported an adequate weight gain. Regarding neonatal characteristics, 6.3% of newborns were preterms, 6.5% had low birth weight, and 4.4% had macrosomia. The median of birth weight was 3.3 kg (IQR = 0.6), and the median of birth length was 50 cm (IQR = 2). Accordingly, 79.2% of newborns were AGA, 10.1% were SGA, and 10.7% were LGA.

### 3.2. Maternal and Neonatal Characteristics According to Socio-Economic Status

As showed in Table 1, we compared women of the MAMI-MED cohort according to their employment status. We found that employed women were slightly older than the unemployed counterpart, with a median age of 32 years compared with 30 years (*p* < 0.001). Moreover, a higher proportion of employed women were primiparous (*p* < 0.001) with a medium–high educational level (*p* < 0.001).

Regarding lifestyles, employed women were more likely to be non-smokers (*p* < 0.001) and to report a medium–high adherence to the MD (*p* < 0.001) and lower pre-pregnancy BMI compared with the unemployed women (*p* < 0.001).

Employed women also had a slightly higher GWG (12.0 kg vs. 11.0 kg, *p* < 0.001). Regarding pregnancy outcomes, employed women had lower proportions of preterm birth (*p* = 0.021) and low-birth weight newborns (*p* = 0.003). Birth weight was significantly higher among infants born to employed women (3.32 kg vs. 3.24 kg; *p* < 0.001). Statistically significant differences were also evident for gestational age at delivery and birth length for newborns between the two groups (*p* < 0.001 and *p* = 0.003, respectively).

Table 2 showed maternal characteristics according to the educational level. Women with medium–high educational levels were more likely to be older (*p* < 0.001), primiparous (*p* < 0.001), and non-smokers (*p* < 0.001) than those with low educational levels. Furthermore, adherence to the MD (*p* = 0.014) and MDS (*p* = 0.003) were higher in the medium–high education group than in the low education one. In addition, women with medium–high education were more likely to follow dietary restrictions than the counterpart (*p* = 0.008). However, no significant difference in total energy intake between the two groups was evident (*p* = 0.380). We also noted a higher proportion of employed women in the medium–high education group (*p* < 0.001). Women belonging to the medium–high education group were more likely to report a lower pre-pregnancy BMI and have normal weight compared with those in the low education group (*p* < 0.001).

Women in the medium–high education group showed a higher GWG (12.0 kg vs. 11.0 kg; *p* = 0.014) and were more likely to have adequate GWG (*p* = 0.010) than the low education group. Regarding pregnancy outcomes, women with medium–high education had a lower proportion of preterm birth (*p* = 0.022). Moreover, there was a significant difference in terms of gestational age at delivery (*p* = 0.013) and birth length (*p* = 0.017). There were no significant differences regarding the proportion of macrosomia and low-birth weight newborns and birth weight by educational level groups.

### 3.3. Characteristics of Clusters

Cluster 1 (n = 739) was characterized by higher proportions of women with lower educational attainment who were unemployed, overweight and/or obese, and had a lower mean MDS. Instead, cluster 2 (n = 773) was mostly characterized by women with a medium–high level of education who were employed, had normal weight, and had a higher average MDS. Thus, we compared the characteristics of women according to their cluster membership (Table 3). Women belonging to cluster 1 were younger (*p* < 0.001) and less likely to be primiparous (*p* < 0.001) than those in cluster 2. In terms of women’s lifestyles, cluster 1 had a lower proportion of non-smokers (*p* < 0.001), lower adherence to the MD (*p* < 0.001), and a slightly lower average of MDS (*p* < 0.001) compared with cluster 2. However, no statistically significant differences were evident between the clusters regarding proportion of women who followed dietary restrictions and total energy intake. Moreover, we found a higher proportion of unemployed women in cluster 1 than in cluster 2 at 90.7% and 8.0%, respectively (*p* < 0.001). Women in cluster 1 had a higher pre-pregnancy BMI (*p* < 0.001) and were more likely to be overweight or obese than those in cluster 2 (*p* < 0.001). GWG was lower in cluster 1 (11.0 kg) compared with cluster 2 (12.0 kg, *p* = 0.017). Compared with cluster 2, women belonging to cluster 1 reported significantly higher proportions of preterm birth (*p* = 0.004) and low-birth weight newborns (*p* = 0.002). Additionally, women in cluster 1 reported higher proportion of LGA newborns than those in cluster 2, although this difference was not statistically significant (*p* = 0.074). Statistically significant differences were also found between the two clusters in terms of week of delivery (*p* < 0.001), birth weight (*p* < 0.001), and length of newborns (*p* = 0.004). However, no significant differences were evident in terms of macrosomia (*p* = 0.352).

## 4. Discussion

Our findings emphasize the critical role of addressing multiple risk factors even before conception and highlight the potential of cluster analysis as a valuable tool for classifying women in accordance with socio-behavioral characteristics. Previous studies have explored the associations between maternal conditions, lifestyle exposures, and embryonic growth [35]. However, most studies to date have primarily examined the impact of individual maternal risk factors on embryonic development and neonatal outcomes in isolation, despite the fact that pregnant women are frequently subjected to a combination of social characteristics, lifestyles, and medical risk factors concurrently [36]. The numerous physiological changes during gestation indicate that the early classification of women is crucial for promptly identifying those at higher risk, so that timely and tailored public health interventions can be implemented [37]. In general, the clustering approach serves as an advanced data mining technique for grouping individuals based on key variables [38]. This statistical method is particularly valuable for enabling the exploration of previously unknown correlations between pathophysiological factors and facilitating the prediction of specific mother–child outcomes [38,39,40]. Interestingly, our findings show the critical need to explore the intricate interplay between socio-demographic and behavioral factors in shaping maternal and infant trajectories risk, highlighting that employment may be considered a key factor in influencing maternal and neonatal health outcomes. Indeed, employed women tend to be slightly older, are more likely to be primiparous, and exhibit higher rates of non-smoking and adherence to the MD. This highlights the protective benefits of employment, such as financial stability, better access to prenatal care and health information, healthier living habits and lifestyles choices, and lower stress levels, all of which contribute to improved maternal health [21]. Furthermore, the significant disparity in educational levels between employed and unemployed women further emphasizes the crucial role of education and health literacy in the peri-conceptional period. Numerous pathways have been proposed to explore the relationship between education and maternal and child health, including skill development, socialization, information dissemination, and delays in childbearing [41]. In line, our study underscores that women with medium–high educational levels, as well as those who are employed, demonstrated healthier behaviors and more favorable pregnancy outcomes. This group exhibited lower pre-pregnancy BMI, higher gestational weight gain, and experienced fewer instances of preterm birth. These findings align with the broader understanding that higher educational attainment and employment enhance health literacy, encourage healthier lifestyle choices, and provide better access to healthcare resources, ultimately contributing to improved maternal and neonatal health outcomes [3]. Maternal educational level is strongly associated with a range of preventive and treatment-related health behaviors, including extended breastfeeding, thorough child care, and the effective use of health services [42]. In general, these health behaviors play a key role in mediating the relationship between a woman’s level of education and the health outcomes for both mothers and their children [43,44]. Although the above-mentioned analyses revealed significant associations, considering maternal social factors in isolation does not fully address their combined impact on long-term mother-child outcomes. For these reasons, we next grouped women based on a comprehensive set of factors, including social characteristics, educational attainment, and lifestyle choices. One cluster comprised women who displayed characteristics of being more cautious, attentive, and well-informed, often reflecting a more proactive approach to health and well-being. Women in cluster one exhibited less favorable socio-economic characteristics, lifestyles, and pregnancy outcomes compared with those in the second cluster. In particular, we found that cluster one consisted of women who may experience greater socio-economic challenges and are less likely to adhere to a healthy dietary pattern. By contrast, the second cluster consists of women with more advantageous socio-economic and healthier lifestyle choices.

## 5. Limitations

Our study has several limitations that should be considered. First, the selection of variables for the cluster analysis was somewhat subjective and may have influenced the clusters we identified. Our analysis did not include additional socio-demographic and behavioral variables that could have revealed different clusters and identified other groups of women. Second, while the clusters show some degree of homogeneity, variations within each cluster still exist. Pregnant women may alter their behaviors upon discovering their pregnancy, and unconsidered confounding factors, such as genetic predispositions or pre-existing health conditions, could also impact the clustering results. Therefore, incorporating longitudinal data in future research could provide insights into how maternal behaviors change during pregnancy. Third, we chose to use the two-step cluster analysis for our study, even though more advanced techniques, such as latent class analysis (LCA), now exist, which could improve our analysis by modeling latent relationships more effectively. However, in the near future, it could be useful to apply LCA to our data and in contexts where latent data structure and prior knowledge of expected patterns are crucial.

Another limitation is the precision in measuring dietary habits. The adherence to the MD and the MDS were based on the FFQ, a self-reported tool. As such, the FFQ is prone to errors and inaccuracies, including recall bias and misreporting, which could affect the reliability of the dietary data collected.

## 6. Conclusions

In conclusion, this study provides a comprehensive understanding of how women with similar socio-demographic characteristics often adopt similar lifestyles, which can impact adverse neonatal outcomes. Our findings highlight that applying a clustering approach during pregnancy and the peri-conceptional period can effectively identify groups of women who might benefit from targeted interventions. These interventions could reduce disparities in education and employment while encouraging healthier lifestyle choices, highlighting the crucial role of educational initiatives and support programs in improving maternal and child health outcomes.

From a policy perspective, this research underscores the importance of designing policies that address the socio-demographic factors linked to maternal and neonatal health risks. Policy initiatives focused on accessible education, employment support, and healthcare access for women, particularly those in socio-economically disadvantaged groups, could contribute to mitigating inequalities in birth outcomes. This would involve not only enhancing healthcare accessibility but also integrating community-based programs that promote the awareness of healthy behaviors during the peri-conceptional period. For health practice, the findings suggest that healthcare providers should consider socio-demographic factors in their assessment of maternal health risk profiles. Implementing a clustering-based approach in clinical settings could aid healthcare professionals in identifying women at higher risk for adverse outcomes, allowing for more precise, individualized intervention strategies. Such strategies might include targeted nutritional counseling, mental health support, and guidance on lifestyle adjustments, which could improve both maternal and neonatal outcomes. However, further prospective research during the peri-conceptional period is needed to fully understand how complex interactions between socio-demographic, behavioral, and genetic factors influence the risk of abnormal outcomes in the first trimester and complications for both mothers and infants. Expanding upon these findings could offer valuable insights to refine both policy interventions and clinical practices, enabling more effective strategies to reduce maternal and neonatal health disparities.

## Figures and Tables

**Table 1 nutrients-16-03975-t001:** Characteristics of the study sample by employment status.

Characteristics	Employed (n = 780)	Not Employed (n = 732)	*p*-Value ^a^
Age (years) ^b^	32.0 (5.0)	30.0 (7.0)	<0.001
Primiparous	64.3%	44.4%	<0.001
Non-smoker	93.4%	86.9%	<0.001
Gestational week at delivery (weeks) ^b^	39.0 (1.0)	39.0 (2.0)	<0.001
Adherence to MD			
Low	33.7%	42.5%	<0.001
Medium–high	66.3%	57.6%	
MDS ^b^	4.0 (2.0)	4.0 (2.0)	<0.001
Dietary restrictions (% yes)	15.8%	14.6%	0.533
Total energy intake (kcal/day) ^b^	1716 (545)	1716 (583)	0.607
Educational level			
Low	8.8%	41.8%	<0.001
Medium–high	91.2%	58.2%	
Pre-pregnancy BMI (kg/m^2^) ^b^	22.7 (5.3)	24.1 (7.0)	<0.001
Pre-pregnancy BMI classification			
Underweight	5.8%	5.6%	<0.001
Normal weight	64.1%	52.0%	
Overweight	19.9%	25.1%	
Obese	10.3%	17.2%	
GWG (kg) ^b^	12.0 (7.0)	11.0 (8.2)	<0.001
GWG classification			
Reduced	37.7%	41.4%	0.345
Adequate	33.1%	31.5%	
Excessive	29.2%	27.1%	
Preterm birth	4.9%	7.8%	0.021
Low birth weight (% yes)	4.6%	8.6%	0.003
Macrosomia (% yes)	4.5%	4.2%	0.802
Birth weight (kg) ^b^	3.32 (0.55)	3.24 (0.55)	<0.001
Birth length (cm) ^b^	50.0 (2.0)	50.0 (2.0)	0.003

^a^ *p*-values are obtained through the chi-squared test for qualitative variables and the Mann–Whitney U-test for quantitative variables. ^b^ Data are reported as the median (IQR). Abbreviations: MD, Mediterranean diet; BMI, body mass index; GWG, gestational weight gain.

**Table 2 nutrients-16-03975-t002:** Characteristics of women of the MAMI-MED cohort according to the educational level.

Characteristics	Low Educational Level (n = 375)	Medium–High Educational Level (n = 1137)	*p*-Value ^a^
Age (years) ^b^	28.0 (8.0)	31.0 (6.0)	<0.001
Primiparous	40.6%	59.3%	<0.001
Non-smoker	80.8%	93.4%	<0.001
Gestational week at delivery (weeks) ^b^	39.0 (2.0)	39.0 (2.0)	0.013
Adherence to MD			
Low	44.0%	36.0%	0.014
Medium–high	56.0%	64.0%	
MDS ^b^	4.0 (2.0)	4.0 (2.0)	0.003
Dietary restrictions (% yes)	10.9%	16.6%	0.008
Total energy intake (kcal/day) ^b^	1730 (608)	1709 (543)	0.380
Employment status			
Employed	18.4%	62.5%	<0.001
Not employed	81.6%	37.5%	
Pre-pregnancy BMI (kg/m^2^) ^b^	24.1 (7.5)	23.1 (5.9)	<0.001
Pre-pregnancy BMI classification			
Underweight	4.3%	6.2%	<0.001
Normal weight	51.7%	60.4%	
Overweight	22.9%	22.3%	
Obese	21.1%	11.2%	
GWG (kg) ^b^	11.0 (9.0)	12.0 (7.0)	0.014
GWG classification			
Reduced	43.6%	38.2%	0.010
Adequate	25.8%	34.4%	
Excessive	30.6%	27.4%	
Preterm birth	8.8%	5.5%	0.022
Low birth weight (% yes)	9.1%	5.7%	0.055
Macrosomia (% yes)	5.1%	4.1%	0.447
Birth weight (kg) ^b^	3.3 (0.54)	3.3 (0.58)	0.209
Birth length (cm) ^b^	50.0 (3.0)	50.0 (2.0)	0.017

^a^ *p*-values are obtained through the chi-squared test for qualitative variables and the Mann–Whitney U-test for quantitative variables. ^b^ Data are reported as the median (IQR). Abbreviations: MD, Mediterranean diet; BMI, body mass index; GWG, gestational weight gain.

**Table 3 nutrients-16-03975-t003:** Characteristics of the study sample by clusters.

Characteristics	Cluster 1 (n = 739)	Cluster 2 (n = 773)	*p*-Value ^a^
Age (years) ^b^	29.0 (7.0)	32.0 (5.0)	<0.001
Primiparous	44.1%	64.8%	<0.001
Non-smoker	85.9%	94.4%	<0.001
Gestational week at delivery (weeks) ^b^	39.0 (2.0)	39.0 (2.0)	<0.001
Adherence to MD			
Low	43.4%	32.7%	<0.001
Medium–high	56.6%	67.3%	
MDS ^b^	4.0 (2.0)	4.0 (2.0)	<0.001
Dietary restrictions (% yes)	13.9%	16.4%	0.177
Total energy intake (kcal/day) ^b^	1715 (579)	1715 (540)	0.909
Employment status			
Employed	9.3%	92.0%	<0.001
Not employed	90.7%	8.0%	
Pre-pregnancy BMI (kg/m^2^) ^b^	23.9 (7.1)	22.8 (5.4)	<0.001
Pre-pregnancy BMI classification			
Underweight	5.5%	5.8%	<0.001
Normal weight	53.2%	63.1%	
Overweight	24.4%	20.6%	
Obese	16.9%	10.5%	
GWG (kg)^b^	11.0 (8.0)	12.0 (6.0)	0.017
GWG classification			
Reduced	40.6%	38.4%	0.619
Adequate	31.2%	33.3%	
Excessive	28.1%	28.2%	
Preterm birth	8.1%	4.6%	0.004
Low birth weight (% yes)	8.7%	4.5%	0.002
Macrosomia (% yes)	4.9%	3.9%	0.352
Birth weight (kg) ^b^	3.2 (0.56)	3.3 (0.58)	<0.001
Birth length (cm) ^b^	50.0 (3.0)	50.0 (2.0)	0.004
Birth weight for gestational age			
SGA	11.6%	8.6%	0.074
AGA	76.9%	81.5%	
LGA	11.5%	9.9%	

^a^ *p*-values are obtained through the chi-squared test for qualitative variables and the Mann–Whitney U-test for quantitative variables. ^b^ Data are reported as the median (IQR). Abbreviations: MD, Mediterranean diet; BMI, body mass index; GWG, gestational weight gain.

## Data Availability

The data presented in this study are available on request from the corresponding author due to privacy restrictions.

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
