# Peer review of "Social and Nutritional Profiles of Pregnant Women: A Cluster Analysis on the “MAMI-MED” Cohort"

_nutrients, 2024, doi:10.3390/nu16233975_

Round 1

Reviewer 1 Report

Comments and Suggestions for Authors

It is an interesting study about the relationship of on the one hand social determinants and dietary habits of pregnant women and on the other hand later maternal and neonatal outcomes in an Italian cohort study. It is a generally well written paper, I only have some minor comments.

In the Methods section (line 100-101) I just wonder why medium educational level is defined as 9-13 years of school and high educational level as >8 years of school. It means that middle level is part of the high educational level. Is it correct?

Results (line 172) please put ‘2’ in upper index (in kg/m2).

Table 1, 2 and 3: In some cases (gestational week at delivery; MDS; birth length) it is strange that although the same number is given for the two groups, there is a statistically significant difference. It would look better if the data were presented to two decimal places, because then it would be possible to see which group had significantly lower values. In their current form these tables don’t support line 188, 193-195, 206, 217-218, 231, 236 and 247-248 statements.

Author Response

Dear Editor,

this document is intended to facilitate the work of the editor and reviewers by presenting a comprehensive list of requested modifications. We are pleased to submit a revised version of the manuscript that incorporates all received comments and suggestions. The following is a detailed list outlining the changes made in response to the reviewers' comments (highlighted in yellow).

Reviewer 1

It is an interesting study about the relationship of on the one hand social determinants and dietary habits of pregnant women and on the other hand later maternal and neonatal outcomes in an Italian cohort study. It is a generally well written paper, I only have some minor comments.

Comment (C): In the Methods section (line 100-101) I just wonder why medium educational level is defined as 9-13 years of school and high educational level as >8 years of school. It means that middle level is part of the high educational level. Is it correct?

Author (A): Thank you for pointing out the potential overlap in the educational level definitions. We have revised the definitions as follows to avoid overlap: low education level (≤8 years) and medium- high >8years.

C: Results (line 172) please put ‘2’ in upper index (in kg/m2).

A: We agree with your suggestion. As requested, we modified the sentence accordingly.

C: Table 1, 2 and 3: In some cases (gestational week at delivery; MDS; birth length) it is strange that although the same number is given for the two groups, there is a statistically significant difference. It would look better if the data were presented to two decimal places, because then it would be possible to see which group had significantly lower values. In their current form these tables don’t support line 188, 193-195, 206, 217-218, 231, 236 and 247-248 statements.

A: Thank you for your insightful comments regarding Tables 1, 2, and 3. We appreciate your observation about the presentation of identical values between the two groups (gestational week at delivery, MDS, birth length) and the presence of statistically significant differences.

Even if we presented values to two decimal places, the medians for these variables would remain the same between groups. The statistical significance arises from the underlying distributional differences detected by non-parametric tests, specifically the Mann-Whitney U test, which evaluates the overall rank distributions rather than comparing median or mean values directly. Therefore, statistical significance can be present even when median values appear identical.

Reviewer 2 Report

Comments and Suggestions for Authors

The authors have undertaken an impressive scope of analysis, and the manuscript largely succeeds. A number of points should be addressed in a revision.

The cluster analysis results look interesting, and the results seem to have been interpreted correctly. It is important to note that latent class analysis (LCA) is a more modern version of cluster analysis and is more likely to produce interesting results. It can be conducted in SPSS, but likely would require a software extension. The authors do not need to re-do the analysis using LCA, but should at minimum indicate that using a dated method is a substantial limitation of the analysis.

The discussion appears to be one very long paragraph, which makes it very difficult to locate the principal ideas expressed in that verbiage. That overly extensive paragraph needs serious editing into more readable chunks. Similar editing throughout the manuscript would provide for greater readability.

The abbreviated Conclusions segment is the opposite situation. It could be expanded easily to include discussion of key implications of the findings for policy initiatives and interventions and of implications for health practice. Doing so would greatly increase the likelihood of the research having broader impact.

Lines 157-159 state: "For categorical variables, the two-sided Chi-squared test was employed, while the Kruskal-Wallis test was used for continuous variables. All statistical tests were two-tailed, with a significance level set at p<0.05." Whether chi-square is a two-tailed test is in the eye of the beholder. Operationally the right tail is all that matters.

Lines 163 196, and 254 should say "study sample" rather than "study population"

Author Response

Dear Editor,

this document is intended to facilitate the work of the editor and reviewers by presenting a comprehensive list of requested modifications. We are pleased to submit a revised version of the manuscript that incorporates all received comments and suggestions. The following is a detailed list outlining the changes made in response to the reviewers' comments (highlighted in yellow).

Reviewer 2

The authors have undertaken an impressive scope of analysis, and the manuscript largely succeeds. A number of points should be addressed in a revision.

C: The cluster analysis results look interesting, and the results seem to have been interpreted correctly. It is important to note that latent class analysis (LCA) is a more modern version of cluster analysis and is more likely to produce interesting results. It can be conducted in SPSS, but likely would require a software extension. The authors do not need to re-do the analysis using LCA, but should at minimum indicate that using a dated method is a substantial limitation of the analysis.

A: We appreciate your suggestion and have addressed it in the limitations section. We acknowledge that Latent Class Analysis (LCA) holds great potential for future research and may be particularly useful in settings where understanding latent data structure and leveraging prior knowledge of expected patterns are critical. However, in our case, LCA has several practical and methodological challenges, including the need for additional software extensions that were not readily available, which would have increased both costs and analysis time. Given the exploratory nature of our study, we opted to use cluster analysis to identify homogeneous groups of women, as this approach aligned with our resource constraints and enabled us to provide more straightforward and interpretable results.

C: The discussion appears to be one very long paragraph, which makes it very difficult to locate the principal ideas expressed in that verbiage. That overly extensive paragraph needs serious editing into more readable chunks. Similar editing throughout the manuscript would provide for greater readability.

A: As requested, we have improved the discussion sections to avoid redundancies and highlight the key contributions, as well as implications, of our work.

C: The abbreviated Conclusions segment is the opposite situation. It could be expanded easily to include discussion of key implications of the findings for policy initiatives and interventions and of implications for health practice. Doing so would greatly increase the likelihood of the research having broader impact.

A: Thank you very much for your valuable feedback regarding the Conclusions section. We agree that expanding this section could strengthen the impact of the research, particularly in terms of its implications for policy initiatives, interventions, and health practice. Accordingly, we revised the Conclusions to include a more detailed discussion on these aspects, highlighting the potential applications of the findings to broader health policies and clinical practices. We appreciate your insight, which has been instrumental in improving the manuscript.

C: Lines 157-159 state: "For categorical variables, the two-sided Chi-squared test was employed, while the Kruskal-Wallis test was used for continuous variables. All statistical tests were two-tailed, with a significance level set at p<0.05." Whether chi-square is a two-tailed test is in the eye of the beholder. Operationally the right tail is all that matters.

A: Thank you for your suggestion. We apologize for the mistake and for the lack of precision in our wording. Thus, we revised the text in lines 157-159 to improve the description of the Chi-squared test.

C: Lines 163 196, and 254 should say "study sample" rather than "study population"

A: Done

Round 2

Reviewer 2 Report

Comments and Suggestions for Authors

The revisions are responsive to reviewer comments.